# MULTI-CLASS FEW SHOT LEARNING TASK AND CONTROLLABLE ENVIRONMENT

## ABSTRACT

Deep learning approaches usually require a large amount of labeled data to generalize. However, humans can learn a new concept only by a few samples. One of the high cogntition human capablities is to learn several concepts at the same time. In this paper, we address the task of classifying multiple objects by seeing only a few samples from each category. To the best of authors' knowledge, there is no dataset specially designed for few-shot multiclass classification. We design a task of mutli-object few class classification and an environment for easy creating controllable datasets for this task. We demonstrate that the proposed dataset is sound using a method which is an extension of prototypical networks.

## 1 INTRODUCTION

Deep learning approaches are usually capable of solving a classification problem when a large labeled dataset is available during the training (Krizhevsky et al., 2012; Sutskever et al., 2014; Deng et al., 2013). However, when a very few samples of a new category is shown to a trained classifier, it either fails to generalize or overfit on the new samples. Humans, however, can easily generalize their prior knowledge to learn a new concept even with one sample. Few-shot learning approaches are proposed to address this gap between human capablity of learning a new concept with only a very few labled samples and machine capablity in generalizing to a new concept. mini-ImageNet (Vinyals et al., 2016) and tiered-Imagenet (Ren et al., 2018) are two main datasets that are developed to help the research community addressing the problem of few-shot classification.
Although that human are capable of learning a new concept with only a very few samples. Learning a few *new* concepts at the same time, and with only a very few samples of each is considered as a high cognition task (Carey & Bartlett, 1978) and very challenging even for humans.

It is yet an active area of study to know how human are capable of doing this. There could be many factors involved in this high cognition process, and there are many hypothesis around this. One popular hypothesis is that the brain is able to learn a *good representation* that has high capacity and can generalize well (Goodfellow et al., 2016). Studying the reasons behind human high cognitive capablity of learning a few new concepts in paralell and with only a very few samples, is out of the scope of this paper. However, in this paper, we propose to extend the few shot learning problem to multi-class few shot classification problem and moving a step towards filling the gap between human cognitive capablity of learning multiple new concepts in paralel and with only a few samples, and machine learning approaches.

To do so, our first step is to define a dataset and a setup to address this problem, and an evaluation metric to measure our progression towards solving this problem.

We argue that the existing datasets are not desirable for this task. Omniglot (Lake et al., 2015), mini-ImageNet, tiered-ImagaNet, are designed for single object classification. Such datasets as, MS COCO (Lin et al., 2014) and Pascal VOC (Everingham et al., 2015) have multiple object classes but they are not well suited for few-shot learning. The issue is the high imbalance of class co-occurrence (for example, 'human' label occures with all other classes). Therefore it is hard to prevent the learner from "sneak peeking" new concepts.

To sum it up, this work's contribution is two-fold:

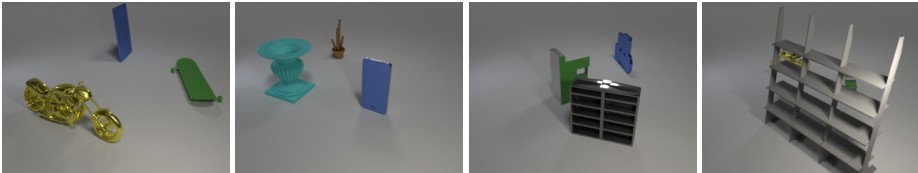

Figure 1: 3-dense samples from the dataset. Right two images demonstrate different degrees of occlusion.

1. We propose the new task of mutli-object few-shot classification to test model's ability to disentagle and represent several object on an image (see Section 3) and propose an extension to prototypical-style models to efficiently solve the task (Section 3.1);

2. We construct a new dataset which provides clean and controlled environment of multi-object images (see Section 4) and provide the framework, benchmarks and the code for the community to explore controlled scenarios and alternative few-shot classification tasks.

## 2 RELATED WORK

The problem of learning new concepts with small number of labeled data is usually referred to as *few-shot learning* (Larochelle, 2018). Two of the most famous datasets to address the problem of few shot classification are mini-Imagenet (Vinyals et al., 2016) and tiered-Imagenet (Ren et al., 2018). Both of these datasets address the problem of few shot classification of single objects.

Kang et al. (2018) addresses the problem of few shot object detection in natural images. There are two main groups of approaches addressing this problem (i) *optimization-based* frameworks and (ii) *metric-based* frameworks. Optimization-based framework (Andrychowicz et al., 2016; Finn et al., 2017; Rusu et al., 2018), is a class of algorithms that learn how to quickly learn new concepts. Other notable performant approaches that do not fall into these two categories is SNAIL (Mishra et al., 2017).

Meta-learning approaches work by learning a parameterized function that maps labeled training sets into classifiers.

Metric-based framework learn a representation that minimize intra-class distances while maximize the distance between variant classes. These approaches usually rely on an episodic training framework: the model is trained with sub-tasks (episodes) in which there are only a few training samples for each category. Matching networks (Vinyals et al., 2016) trains a similarity function between images. In each episode, it uses an attention mechanism (over the encoded support) as a similarity measure for one-shot classification.

In prototypical networks (Snell et al., 2017), a metric space is learned where embeddings of queries of one category are close to the centroid (or prototype) of support of the same category, and far away from centroids of other classes in the episode. Due to simplicity and performance of this approach, many methods extended this work. For instance, Ren et al. (2018) propose a semi-supervised few-shot learning approach and show that leveraging unlabeled samples outperform purely supervised prototypical networks. Wang et al. (2018) propose to augment the support set by generating hallucinated examples. Task-dependent adaptive metric (TADAM) (Oreshkin et al., 2018) relies on conditional batch normalization (Perez et al., 2018) to provide task adaptation (based on task representations encoded by visual features) to learn a task-dependent metric space.

## 3 MULTI-OBJECT FEW-SHOT CLASSIFICATION TASK

In order to test the ability to disentangle unseen objects on a given image, we propose a task of multi-object few-shot classification.

**Few-shot classification** First, we briefly summarize the single object few-shot classification (Larochelle, 2018). While a dataset for supervised learning is comprised of a number of input-

label pairs, a dataset $D$ for meta-learning is comprised of a number of tasks. In a common $K$-shot $N$-way classification every task has a support set $S = \{(x_i, y_i)\}_i$ of the size $KN$ and a query set $Q = \{(x_i, y_i)\}_i$. The learner is given a number of support-query pairs for training. For testing, the learner is given am unseen support set $S$ and asked to predict labels for a certain query $Q = x$, which label is one of the labels in the support. Usually, prototypes are computed in the following way. First, the images from the support set $e_i = \mathrm{CNN}(x_i)$ are embedded, the prototype for each class is computed as an average embeding $p_n = \mathrm{mean}_{i:y_i=n} e_i$ for all $n \in 1 \ldots N$.

**Multi-object classification**  In this work, we propose to extend this task for a multi category case. We define a $K$-shot $N$-way $D$-dense task to have the support set $S = \{(x_i, y_i^d)\}_i$ of the size $\frac{KN}{D}$ where every image $x_i$ contains $D$ objects and $y_i = (y_i^1 \ldots y_i^D)$ is a tuple of $D$ labels corresponding the objects. This way, the learner is exposed to every object class exactly $K$ times, therefore our methods can be easier compared among varying $D$. Similarly, the query $Q = \{(x_i, Y_i)\}_i$ is a set of images each containing $D$ objects with ground truth labels $y_i$.

## 3.1  Multi-Prototype Networks

A naïve aproach requires exponential in $D$ pseudo-label to represent all possible combinations. The learner can only be exposed to a limited number of possible combinations of objects. The exponential size of the label quickly overpasses few shots (commonly from 5 to 20) available for training.

We propose *multi-prototype networks* to tackle aforementioned exponential explosion. For simplicity, in this work we assume that the labels $y_i$ are given in the order that objects appear on the image $x_i$ from left to right. This is a major simplification and we will lift it in the near future work. To extend the proto-net setup, we train a model to produce $D$ embeddings for every image $e_i^d = \mathrm{CNN}(x_i)$. The prototype is computed as the average embedding per class $p_n = \mathrm{mean}_{i,d:y_i^d=n} e_i^d$. The rest of the procedure is identical to the proto-net – every query is compared to the prototypes and the distance to the correct prototype is pushed down, all the rest are pushed up.

## 4  CEMOL: Controlled Environment for Multiple Object Learning

We aim to have a controlled environment for reliable experiments. To achieve this, we develop a dataset based on Shapenet 3D models renderred with Blender in the setup similar to CLEVR. This provides us flexibility to construct single or multiple object tasks and change the task parameters – number of shots and ways. The dataset along with the generation tools will be made publically available.

In the next sections we describe in the detail the dataset generation procedure. Then, to compare the complexity of the dataset to existing ones, we run a popular model TADAM (Oreshkin et al., 2018) for a traditional single-object-per-image setup. Then, we increase the number of objects per image and report results with our proposed model.

## 4.1  Dataset Generation Details

We generated a dataset using methods similarly to visual reasoning dataset CLEVR (Johnson et al., 2017). To increase the variability of the object classes, we used 3D meshes from Shapenet (Chang et al., 2015) dataset. Beforehand, we splitted 55 Shapenet classes into three subsets randomly for training, validation and testing (see Appendix A). The sizes of the subsets are 35, 10, 10, respectively. We render images task-by-task. We uniformly sample classes used in the task. Then, we sample a mesh of each class $K$ times. We distribute object over canvases. Finally, we render images by randomly placing meshes on each canvas. In order to produce the representation of an object independent of color, we strip the texture and apply random color random texture of either rubber or metal. Figure 2 demonstrates some samples from the test split. The model should be able to tackle

Table 1: Comparison of accuracy of the most prominent models on mini-ImageNet. Single label version of our dataset. All accuracies reported with 5-way task.

| Dataset | 1-shot | 5-shot |
|---|---|---|
| **miniImageNet** | | |
| ProtoNet (Snell et al., 2017) | 49.4% | 68.2% |
| TADAM (Oreshkin et al., 2018) | 58.5% | 76.7% |
| LEO (Rusu et al., 2018) | 61.8% | 77.6% |
| **CEMOL** | | |
| resnet12 + protonet | 63.6% | 64.8% |
| resnet12 + TADAM | 78.0% | 80.6% |

Table 2: Comparison of proto-net performance with various density (number of objects per image) for 5-shot 6-way task.

| Density | Accuracy, % |
|---|---|
| 1 | $68.6 \pm 0.5$ |
| 2 | $51.3 \pm 0.3$ |
| 3 | $48.2 \pm 0.2$ |

complex shapes as well as occlusions. We renderred 110,000 images for training, 35,000 for each validation and testing splits and we provide the source code to generate extra data of any density.[1]

## 4.2 Comparison to Existing Datasets

First, we generate only images containing single object in order to place it among the exising ones. We train a prototypical network and TADAM with 12-block residual feature extractor (He et al., 2016). We summarize our results in the Table 1 and compare to performance on mini-ImageNet. As it can be seen, our dataset is simpler than mini-ImageNet, but it cannot be trivially solved by close to state-of-the-art methods even in the simplest single-object case. Therefore, we conclude that the dataset is not trivial nor too hard.

## 4.3 Multiple Objects

Having shown that the proposed dataset is sound for single-object task, we increase the number of objects per image and apply the proposed MultiProtoNet (see Section 3.1). In all experiments below, we used 12-block residual network that produces a corresponding number of embeddings per image. All networks are optimized with Adam (Kingma & Ba, 2014) with the Euclidian distance metric.

The experiments are summarized in the Table 2. We notice that while the accuracy accuracy drops significantly when transitioning from single to multiple objects, it drop as much from 2 to 3 objects.

## 5 Discussion and Future Work

In this work we introduced a task of few-shot multi-object classification and an environment for generating datasets for this task. We compared the proposed dataset to existing ones in single-object case. Then, we used a simple extension of prototypical networks to conduct experiments multi-object case. We believe that this task will help diagnosing metric-learning models that need to disentangle several objects on an image.

One of the future directions we are taking is to lift the limitation of known object order (Section 3.1). Then we plan to use stronger feature extractors (Oreshkin et al., 2018) and extend the work to more natural data.

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

## A    TRAIN-TEST SPLIT

Train split: 'table', 'train', 'earphone', 'birdhouse', 'remote control', 'display', 'ashcan', 'car', 'lamp', 'camera', 'faucet', 'bottle', 'bench', 'washer', 'mug', 'microphone', 'knife', 'mailbox', 'rocket', 'guitar', 'sofa', 'laptop', 'basket', 'computer keyboard', 'loudspeaker', 'dishwasher', 'piano', 'rifle', 'microwave', 'bus', 'bowl', 'file', 'pillow', 'cabinet', 'helmet'

Valid split: 'bed', 'cellular telephone', 'vessel', 'bathtub', 'pistol', 'bag', 'airplane', 'chair', 'jar', 'can'

Test split: 'stove', 'clock', 'telephone', 'tower', 'cap', 'skateboard', 'bookshelf', 'motorcycle', 'pot', 'printer'

## B    T-SNE EMBEDDINGS

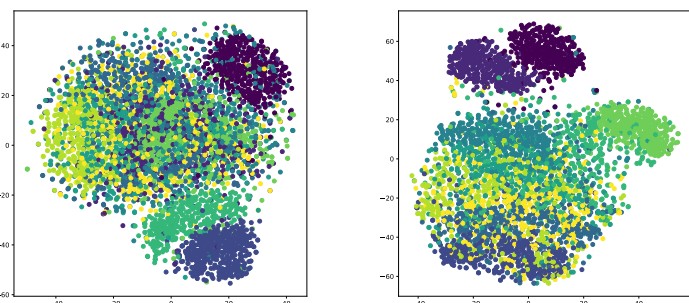

Figure 2: T-SNE embeddings for 3-dense (left) and 2-dense (right) prototypes. Smaller density allows better separation.

