# OpenReview forum: "Multi-Class Few Shot Learning Task and Controllable Environment"
_ICLR.cc/2019/Workshop/LLD — Submitted to LLD 2019_

### Official Review · AnonReviewer2 · 2019-04-02
**needs better justification for treating multiclass settings as a distinct new problem**

**Rating:** 1
**Confidence:** 2

**Review:**

This paper presents a new data set for few shot learning with multiple classes being introduced simultaneously in each example. It also introduces a slight variant on existing few shot architectures first simultaneously learning multiple new classes.

I am unconvinced that the problem of multi-class few shot learning poses a substantially different challenge from single class few shot learning. The cited source, Carey & Bartlett, is a paper entirely about one class few shot learning, so I don't see any evidence that humans, at least, have a distinct method of learning multiple classes simultaneously. The claim, "It is yet an active area of study to know how human are capable of doing this," clearly requires at least one example of active research. While the community would no doubt welcome a new data set for few shot learning, the burden is on the authors to explain why this particular variation on the problem poses a distinct challenge or could be useful to treat separately.

I don't see the significant innovation present in multi-prototype networks. Please contrast them with existing methods for identifying what object in a scene might be referred to with a new label, without trying to simultaneously label multiple objects.

 Minor issues:

Please don't cite an entire textbook (Goodfellow) as a source for a specific claim about human cognition.

This work could benefit greatly from a confident proofreader. Pervasive English grammar and spelling errors make the paper a lot less readable.

---

### Official Review · AnonReviewer1 · 2019-04-10
**A new dataset and task with insufficient empirical support**

**Rating:** 2
**Confidence:** 1

**Review:**

In this paper, the authors defined a new task on multi-object few shot learning. For this new task, the authors presented a dataset generation pipeline as well as a baseline model for future research on this topic. In the experiments, the authors demonstrates 2 claims:

1. The presented data generation pipeline is in a controlled environment. It is neither too hard or too easy, thus can be a good benchmark for future research purpose.

2. The new situation for multi-object few shot learning is observably harder than single object few shot learning.

Comments:

1. Clarity: The clarity of the paper can be substantially improved. It should be written in a more self contained way. E.g. as an educated person without being an expert in this specific topic, it is not crystal clear to me how the protonet works from the description at the beginning of section 3.

2. Significance: This paper proposed multi-object few shot learning. The major difference is the label space is exponential in the number of objects. The distinction from a single object problem with large # of classes would be the exponentially large class combination space. However in Table 1 and 2, the task is designed to have a relatively small # of total class combinations for multi objects. E.g. in table 2, the class combination space size is only 6^3 which is not too large. To this end, the validation of the new task or dataset is not convincing enough to me.

3 Minor: Typo on repeated "accuracy" in the second paragraph of sec 4.3.

---

### Decision · Program_Chairs · 2019-04-10
**Acceptance Decision**

Reject